# Understand the dynamics of GANs via Primal-Dual Optimization

## Abstract

Generative adversarial network (GAN) is one of the best known unsupervised learning techniques these days due to its superior ability to learn data distributions. In spite of its great success in applications, GAN is known to be notoriously hard to train. The tremendous amount of time it takes to run the training algorithm and its sensitivity to hyper-parameter tuning have been haunting researchers in this area. To resolve these issues, we need to first understand how GANs work. Herein, we take a step toward this direction by examining the dynamics of GANs. We relate a large class of GANs including the Wasserstein GANs to max-min optimization problems with the coupling term being linear over the discriminator. By developing new primal-dual optimization tools, we show that, with a proper stepsize choice, the widely used first-order iterative algorithm in training GANs would in fact converge to a stationary solution with a sublinear rate. The same framework also applies to multi-task learning and distributional robust learning problems. We verify our analysis on numerical examples with both synthetic and real data sets. We hope our analysis shed light on future studies on the theoretical properties of relevant machine learning problems.

## 1 Introduction

Since it was first invented by Ian Goodfellow in his seminal work Goodfellow et al. (2014), generative adversarial networks (GANs) have been considered as one of the greatest discoveries in machine learning community. It is an extremely powerful tool to estimate data distributions and generate realistic samples. To train its implicit generative model, GAN uses a discriminator since traditional Bayesian methods that require analytic density functions are no longer applicable. This novel approach inspired by zero sum game theory leads to a significant performance boost; GANs are able to generate samples in a fidelity level that is way beyond traditional Bayesian methods. During the last few years, there have been numerous research articles in this area aiming at improving its performance (Radford et al., 2015; Zhao et al., 2016; Nowozin et al., 2016; Arjovsky et al., 2017; Mao et al., 2017). GANs have now become one of most recognized unsupervised learning techniques and have been widely used in a variety of domains such as image generation (Nguyen et al., 2017), image super resolution (Ledig et al., 2017), imitation learning (Ho & Ermon, 2016).

Despite the great progress of GANs, many essential problems remain unsolved. Why is GAN so hard to train? How to tune the hyper-parameters to reduce instability in GAN training? How to eliminate mode collapse and fake images that show up frequently in training (Arjovsky & Bottou, 2017)? Comparing with many other machine learning techniques, the properties of GANs are far from being well understood. It is quite likely that the theoretical foundation of GANs will become a longstanding problem. The theoretical difficulty of GANs mainly lies in the following several aspects. First, it is a non-convex optimization problem with a complicated landscape. It is unclear how to solve such optimization problems efficiently. The first-order method widely used in the literature via updating the generator and discriminator along descent/ascent direction does not seem to converge all the time. Although some techniques were proposed to stabilize the training performance of the network, e.g., spectral normalization Miyato et al. (2018), in fact, there is no evidence that these algorithms guarantee even local optimality. Second, even if there were an efficient algorithm to solve this optimization problem, we do not know how well they generalize. After all, the optimization formulation is based only on the samples generated by the underlying distribution but our goal is to recover this underlying distribution. Of course, this is a problem faced by all machine learning

techniques. Last, there are no reliable ways to evaluate the quality of trained models. There are a number of works in this topic (Salimans et al., 2016; Heusel et al., 2017), but human eyes inspection remains the primary approach to judge a GAN model.

In the present work, we focus on the first problem and analyze the dynamics of GANs from an optimization point of view. More precisely, we study the convergence properties of the first-order method in GAN training. Our contributions can be summarized as follows. 1) We formulate a large class of GAN problems as a primal-dual optimization problem with a coupling term that is linear over discriminator (see Section 2 for the exact formulation); 2) We prove that the simple primal-dual first-order algorithm converges to a stationary solution with a sublinear convergent rate $\mathcal{O}(1/t)$.

There have been a number of papers that study the dynamics of GANs from an optimization viewpoint. These works can be roughly divided into three categories. In the first category, the authors focus on high level idea using nonparametric models. This includes the original GAN paper Goodfellow et al. (2014), the Wasserstein GAN papers Arjovsky & Bottou (2017); Arjovsky et al. (2017) and many other works proposing new GAN structures. In the second category, the authors consider the unrolled dynamics (Metz et al., 2016), that is, the discriminator remains optimal or *almost* optimal during the optimization processes. This is considerably different to the first-order iterative algorithm widely used in GAN training. Recent works Heusel et al. (2017); Li et al. (2017); Sanjabi et al. (2018) provide global convergence analysis for this algorithm.

The last category is on the first-order primal-dual algorithm, in which both the discriminator and the generator update via (stochastic) gradient descent. However, most of the convergence analysis are local (Daskalakis et al., 2017; Mescheder et al., 2017; Nagarajan & Kolter, 2017; Li et al., 2018). Other related work including the following: In Qian et al. (2018) the authors consider a gradient descent/ascent algorithm for a special min-max problem arising from robust learning (min problem is unconstrained, max problem has simplex constraints); In Yadav et al. (2018) the GANs are treated as convex-concave primal-dual optimization problems. This formulation is considerably different to our setup where GANs, as they should be, are formulated as nonconvex saddle point problems. In Farnia & Tse (2018), the authors investigated the properties of the optimal solutions, which is also different from our work focusing on convergence analysis of the first-order primal-dual algorithm. In Zhao et al. (2018), some unified framework covering several generative models, e.g., VAE, infoGAN, were proposed in the Lagrangian framework. However, the dual variable in their problem is a Lagrangian multiplier, while in our problem, it is the discriminator of GAN. Besides, the focus of their paper is not the optimization algorithm. In Chen et al. (2018), the authors related a class of GANs to constrained convex optimization problems. More specifically, such GANs can be viewed as Lagrangian forms of these convex optimization problems. The optimization variables in their formulation are the probability density of the generator and the function values of the discriminator. Many issues like nonconvexity do not show up. This is essentially a nonparametric model, which doesn't apply to cases when the discriminator and the generator are represented by parametric models. On the other hand, our analysis is carried out on the parametric models directly and we have to deal with the nonconvexity of neural networks. In Hajinezhad & Hong (2017) a primal-dual algorithm has been studied for a non-convex linearly constrained problem (which can be reformulated into a min-max problem, with the max problem being linear and unconstrained, and with linear coupling between variables); In Hamedani & Aybat (2018), (Chen et al., 2013) and the references therein, first-order methods have been developed for convex-concave saddle point problems. Compared to these works, our considered problem is more general, allowing non-convexity and non-smoothness in the objective, non-convex coupling between variables, and can further include constraints. Moreover, we provide global convergence rate analysis, which is much stronger than the local analysis mentioned above.

It turns out that the primal-dual framework we study in this paper can also be applied to the distributional robust machine learning problems (Namkoong & Duchi, 2016) and the multi-task learning problems (Qian et al., 2018). In multi-task learning, the goal is to train a single neural network that would work for several different machine learning tasks. Similarly, in distributional robust learning, the purpose is to have a single model that would work for a set of data distributions. In both problems, an adversarial layer is utilized to improve the worst case performance, which leads to a primal-dual optimization structure that falls into the scope of problems we consider.

The rest of the paper is structured as follows. In Section 2 we introduce GAN and its primal-dual formulation. We provide details of the algorithms with proof sketches in Section 3. The full proofs

are relegated to the appendix. We highlight our theoretical results in Section 4 via several numerical examples, with both synthetic and real datasets.

## 2 GENERATIVE ADVERSARIAL NETWORKS AND MIN-MAX PROBLEMS

GAN is a type of deep generative model with implicit density functions. It consists of two critical components: a generator and a discriminator. The generator takes random variables with known distribution as input and outputs fake samples. The discriminator is trained to distinguish real samples and fake samples.

### 2.1 GANs

In the original form of GAN, the generator is a neural network and the discriminator is a standard classifier with cross entropy cost. Denote the underlying probability distributions of the data sample $\{y_i\}_{i=1}^{n_y} \subset \mathbb{R}^d$ and the random seed $\mathbf{x}$ (a random variable in $\mathbb{R}^k$) of the generative model by $\boldsymbol{P_y}$ and $\boldsymbol{P_x}$ respectively. The original or vanilla GAN is of the form (Goodfellow et al., 2014)

$$\min_{G \in \mathcal{G}} \max_{F \in \mathcal{F}} \{\boldsymbol{E}(\log(F(\mathbf{y}))) - \boldsymbol{E}(\log(1 - F(G(\mathbf{x}))))\}. \tag{1}$$

Here $\mathcal{G}$ stands for the set of possible generators and $\mathcal{F}$ the set of possible discriminators.

There have been numerous modifications of the original GAN, among which the Wasserstein GAN (Arjovsky et al., 2017) attracts a lot of attention. It has the form

$$\min_{G \in \mathcal{G}} \max_{F \in \mathcal{F}} \{\boldsymbol{E}(F(\mathbf{y})) - \boldsymbol{E}(F(G(\mathbf{x})))\}, \tag{2}$$

where $\mathcal{G}$ is the set of parameterized generators and $\mathcal{F}$ is the set of Lipschitz functions with Lipschitz constant 1. This is a special case of a more general class of GANs with the same structure but different constraint set $\mathcal{F}$. The inner maximization loop defines different notions of integral probability metrics (Müller, 1997) depending on the choices of $\mathcal{F}$. Other than Wasserstein GAN, one interesting example with this structure is the generative moment matching networks (Li et al., 2015; Arjovsky et al., 2017). Wasserstein GAN can be extended to general optimal transport cost $c(x, y)$, which results in saddle point formulation

$$\min_{G \in \mathcal{G}} \max_{\psi, \phi} \{\boldsymbol{E}(\psi(\mathbf{y})) + \boldsymbol{E}(\phi(G(\mathbf{x})) \mid \psi(y) + \phi(x) \leq c(x, y), \ \forall x, y\}. \tag{3}$$

Note that the discriminator now becomes two functions $\psi, \phi$ instead of one. When $c(x, y) = \|x - y\|$, the above reduces to the standard Wasserstein GAN in equation 2.

### 2.2 NONCONVEX MIN-MAX PROBLEMS

We observe that, compared to vanilla GAN in equation 1, the Wasserstein GAN in equation 2 and equation 3 has a special structure. In the nonparametric form, the coupling between the generator $G$ and the discriminator $F$ is linear on $F$ in Wasserstein GAN while nonlinear in vanilla GAN. Indeed, replacing $F$ by $\alpha F_1 + \beta F_2$ in the coupling term $\boldsymbol{E}(F(G(\mathbf{x})))$ yields

$$\boldsymbol{E}(F(G(\mathbf{x}))) = \boldsymbol{E}((\alpha F_1 + \beta F_2)(G(\mathbf{x}))) = \alpha \boldsymbol{E}(F_1(G(\mathbf{x}))) + \beta \boldsymbol{E}(F_2(G(\mathbf{x}))).$$

This structure motivates us to study the following min-max primal-dual optimization problem

$$\min_{X \in \mathcal{X}} \max_{Y \in \mathcal{Y}} \quad h(X) + \langle g(X), Y \rangle - l(Y) \tag{4}$$

with $l$ being strictly convex. the other two functions $h$ and $g$ can be non-convex. Here $Y$ represents the discriminator $F$ and $X$ represents the generator $G$. Note that the coupling term $\langle g(X), Y \rangle$ is linear over the discriminator $Y$, but nonlinear in $X$. In real applications, we need to parameterize the discriminator and generator, which may lead to the loss of the property that the coupling is linear over discriminator. Next, we present several cases where this linear structure stays.

### 2.2.1 WASSERSTEIN GAN IN LQG SETTING

Wasserstein GAN in linear quadratic Gaussian (LQG) setting was proposed in Feizi et al. (2017) to understand GAN. In this simplified case, the data distribution is Gaussian, the cost is quadratic and the generator is linear, namely,

$$c(x,y) = \frac{1}{2}\|x - y\|^2, \ \mathbf{y} \sim \mathcal{N}(0, \Sigma_{\mathbf{y}}), \ \mathbf{x} \sim \mathcal{N}(0, I_k), \ G(x) = \theta x.$$

It can be shown that it suffice to consider discriminator of the form

$$\phi(x) = \frac{1}{2}\|x\|^2 - \frac{1}{2}x^T A x, \ \ \psi(y) = \frac{1}{2}\|y\|^2 - \frac{1}{2}y^T B y$$

with $A, B$ being positive definite.

The constraint $\phi(x) + \psi(y) \leq \frac{1}{2}\|x - y\|^2$ implies that $B \succeq A^{-1}$. Consequently, the discriminator can be parametrized by a single variable $A \succ 0$. Additionally, $G(\mathbf{x})$ is a zero mean Gaussian random variable with covariance $\theta\theta^T$. Therefore, the Wasserstein GAN in equation 3 then reduces to

$$\min_{\theta \in \mathbb{R}^{d \times k}} \max_{A \succ 0} \mathrm{Tr}(\Sigma_{\mathbf{y}}) + \mathrm{Tr}(\theta\theta^T) - \mathrm{Tr}(A\theta\theta^T) - \mathrm{Tr}(A^{-1}\Sigma_{\mathbf{y}}), \tag{5}$$

which is apparently in the form of equation 4. When only samples $\{x_i\}_{i=1}^n, \{y_i\}_{i=1}^n$ of $\mathbf{x}, \mathbf{y}$ are available, this becomes

$$\min_{\theta \in \mathbb{R}^{d \times k}} \max_{A \succ 0} \frac{1}{n}\sum_{i=1}^n (\frac{1}{2}\|y_i\|^2 - \frac{1}{2}y_i^T A^{-1} y_i) + \frac{1}{n}\sum_{i=1}^n (\frac{1}{2}\|\theta x_i\|^2 - \frac{1}{2}x_i^T \theta^T A \theta x_i). \tag{6}$$

### 2.2.2 GANS WITH DISCRIMINATORS LINEAR ON FEATURES

In general quadratic discriminators are not sufficient to distinguish complicate high dimension distributions. In order to deal with more general data sets, we consider the setting where the discriminator is a linear combination of predefined basis functions. More specifically, let $\{F_i\}_{i=1}^n$ be the basis functions, and $F = \sum \alpha_i F_i$ with some constraint on $\alpha$, then the above formulation becomes

$$\min_{G \in \mathcal{G}} \max_{\alpha \in \mathbb{R}^n} \left\{ \sum_{i=1}^n \alpha_i \boldsymbol{E}(F_i(\mathbf{y})) - \sum_{i=1}^n \alpha_i \boldsymbol{E}(F_i(G(\mathbf{x})) - \lambda\|\alpha\|^2 \right\}. \tag{7}$$

Here the term $\lambda\|\alpha\|^2$ with $\lambda > 0$ is used to regularize $\alpha$, or equivalently, the discriminator.

Similarly, for GAN structure in equation 3, we can restrict our discriminators $\phi$ and $\psi$ to be linear combinations of basis functions $\{\phi_i\}_{i=1}^n$ and $\{\psi_i\}_{i=1}^n$, that is, $\phi = \sum \alpha_i \phi_i, \ \psi = \sum \beta_i \psi_i$. The constraint $\phi(x) + \psi(y) \leq c(x, y)$ in equation 3 is difficult to impose precisely. Instead, we use a regularization term $l$ and obtain

$$\min_{G \in \mathcal{G}} \max_{\alpha, \beta} \left\{ \sum_i^n \alpha_i \boldsymbol{E}\{\phi_i(G(\mathbf{x}))\} + \sum_i^n \beta_i \boldsymbol{E}\{\psi_i(\mathbf{y})\} - l(\alpha, \beta) \right\}. \tag{8}$$

Clearly, both equation 7 and equation 8 are of the form equation 4. We remark that no constraint has been imposed on the generator $G$; it can be any general neural network. The requirement that the discriminator is a linear combination of predefined basis functions could be strong, but in principle, any function can be approximated to an arbitrary precision with large enough bases.

## 3 ALGORITHM DESIGN AND CONVERGENCE ANALYSIS

In this work, we consider the following general min-max problem,

$$\min_{X \in \mathcal{X}} \max_{Y \in \mathcal{Y}} \quad f(X, Y) \triangleq \frac{1}{n}\sum_{i=1}^n (h_i(X) + \langle g_i(X), Y \rangle - l_i(Y)) \tag{9}$$

where $\mathcal{X}$ is a convex and compact set and the size of $\|X\|$ is upper bounded by $\sigma_X$; $h_i(X) : \mathbb{R}^d \to \mathbb{R}, \forall i$ is a non-convex function and has Lipschitz continuous gradient with constant $L_X$;

$l_i(Y) : \mathbb{R}^d \to \mathbb{R}, \forall i$ are strongly convex with modulus $\gamma > 0$ and Lipschitz gradient constant $L_Y$; the matrix function $g_i(X) : \mathcal{X} \to \dim(Y)$ can also be non-convex, and it is assumed to be Lipschitz and Lipschitz gradient continuous with constants $L_{g,1}$ and $L_{g,2}$. We note that regardless of whether $\mathcal{Y}$ is a bounded set or not, one can show that for all $x \in \mathcal{X}$, the maximizer $Y^*$ for the maximization problem lies in a bounded set; see Lemma 3 in the appendix for proof.

We note that by allowing constraints in the form of $x \in \mathcal{X}$, and $y \in \mathcal{Y}$, one can also include nonsmooth regularizers in the formulation. As an example, if we add $\lambda \|X\|_1$ into the objective function, one can introduce a new variable $z$, and consider an equivalent problem with the constraints $\|X\|_1 \le z$, with the objective function changed to $\frac{1}{n} \sum_{i=1}^{n} (h_i(X) + \langle g_i(X), Y \rangle - l_i(Y)) + z$.

The above formulation is quite general. Compared with the existing convex-concave saddle point literature (Hamedani & Aybat, 2018), (Chen et al., 2013), our formulation allows non-convexity in the minimization, which is essential to modelling the neural network structure of generators in GANs and general non-convex supervised tasks in multi-task learning; Compared with the non-convex linearly constrained problems considered in (Hajinezhad & Hong, 2017), it further allows *non-linear and non-convex* function $g(X)$ to couple with $Y$. Compared with the robust learning formulation given in equation 18, equation 9 can further include constraints and nonsmooth objective functions (thus can include nonsmooth regularizers such as $\ell_1$ norm).

It is important to note that due to the generality of problem equation 9, developing performance guaranteed first-order method, which only utilizing the gradient information about functions $h_i, g_i, l_i$ is very challenging. To the best of our knowledge, there has been no such algorithm that can provably compute even first-order stationary solutions for problem equation 9.

### 3.1 Algorithm description

Our proposed *gradient primal-dual algorithm* of solving equation 9 is listed below, in which we alternatingly perform first-order optimization to update $X$ and $Y$:

$$X^{r+1} = \arg \min_{X \in \mathcal{X}} \left\langle \frac{1}{n} \sum_{i=1}^{n} (\nabla_X h_i(X^r) + \nabla_X \operatorname{Tr}(g_i(X^r)Y^r)), X - X^r \right\rangle + \frac{\beta}{2} \|X - X^r\|^2,$$
$$\tag{10}$$
$$Y^{r+1} = \arg \max_{Y \in \mathcal{Y}} \left\langle -\frac{1}{n} \sum_{i=1}^{n} \nabla_Y l_i(Y^r) + g_i(X^{r+1}), Y - Y^r \right\rangle - \frac{1}{2\rho} \|Y - Y^r\|^2.$$

To be consistent with the optimization literature, we will refer to the $X$-step the "primal step" and the $Y$-step as the "dual step". We note that $1/\beta$ and $\rho$ are two positive parameters, that represent stepsizes of the two updates, and both of them should be small. A few remarks are ready.

**Remark 1 (projected gradient).** It can be easily verified that the updates of $X^r$ and $Y^r$ can be written down in closed form using the following alternating *projected* gradient descent/ascent steps:

$$X^{r+1} = \operatorname{proj}_{\mathcal{X}} \left( X^r - \frac{1}{\beta} \left( \frac{1}{n} \sum_{i=1}^{n} (\nabla_X h_i(X^r) + \nabla_X \operatorname{Tr}(g_i(X^r)Y^r)) \right) \right),$$

$$Y^{r+1} = \operatorname{proj}_{\mathcal{Y}} \left( Y^r + \rho \left( \frac{1}{n} \sum_{i=1}^{n} \nabla_Y l_i(Y^r) + g_i(X^{r+1}) \right) \right).$$

**Remark 2 (stochastic vs deterministic algorithm).** This work will be focused on the *deterministic* algorithm given in equation 10, because such an algorithm is representative of the primal-dual first-order dynamics used in training GANs and optimizing robust ML problems, and it is already challenging to analyze. However, we do want to remark that, it is relatively straightforward to build upon our proof, by incorporating the standard technique in stochastic constrained optimization (Ghadimi et al., 2016) (such as using decreasing stepsizes, and certain randomization rule in picking the final solutions), to analyze the *stochastic* version of the algorithm, in which mini-batches of the component functions are randomly selected to update at each iteration. However, in order to keep the discussion of the paper simple, we choose not to present such results.

### 3.2 CONVERGENCE ANALYSIS

In this section, we present our main convergence results for the primal-dual first-order algorithm given in equation 10. We first present a few necessary lemmas.

**Lemma 1.** *(Descent Lemma) Let $(X^r, Y^r)$ be a sequence generated by equation 10. The descent of the objective function can be quantified by*

$$f(X^{r+1}, Y^{r+1}) - f(X^r, Y^r) \leq -\left(\frac{\beta}{2} - \frac{L_X}{2}\right) \|X^{r+1} - X^r\|^2$$

$$+ \frac{1}{2}\left(L_Y + L_{g,1} + \frac{1}{\rho}\right) \|Y^{r+1} - Y^r\|^2 + \frac{L_X + \frac{1}{\rho}}{2} \|Y^r - Y^{r-1}\|^2. \quad (11)$$

From Lemma 1, it is not clear whether the objective function is decreased or not, since the primal step will consistently decrease the objective value while the dual step will increase the objective value.

The key in our analysis is to identify a proper "potential function", which can capture the essential dynamics of the algorithm, and will be able to reduce in *all iterations*.

**Lemma 2.** *When the following conditions are satisfied,*

$$\rho < \frac{\gamma}{L_Y^2}, \quad \beta \geq \max\left\{L_{g,1} + \frac{(L_Y + L_{g,1} + \frac{1}{\rho})((6\sigma_Y + 1)L_X + L_{g,2} + 2\rho L_{g,1}^2)}{\gamma - \rho L_Y^2}, L_X + \sigma_Y L_{g,2}\right\} \quad (12)$$

*then there exist $c_1, c_2, c_3, \underline{d} > 0$ such that potential function will monotonically decrease, i.e.,*

$$\mathcal{P}^{r+1} - \mathcal{P}^r \leq -c_1\|X^{r+1} - X^r\|^2 - c_2\|Y^{r+1} - Y^r\|^2 - c_3\|Y^r - Y^{r-1}\|^2 \quad (13)$$

*where $\mathcal{P}^{r+1} \triangleq f(X^{r+1}, Y^{r+1}) + \underline{d}\mathcal{Q}^{r+1}$ and*

$$\mathcal{Q}^{r+1} \triangleq \left(\frac{L_X(\sigma_Y + 1)}{2\beta} + \frac{1}{2}\right) \|X^{r+1} - X^r\|^2 + \left(\frac{1}{2\rho\beta} - \frac{\gamma - \rho L_Y^2}{2\beta}\right) \|Y^{r+1} - Y^r\|^2.$$

To state our main result, let us define the proximal gradient of the objective function as

$$\nabla\mathcal{L}(X, Y) \triangleq \left[ \begin{array}{c} X - \text{proj}_{\mathcal{X}}[X - \nabla_X f(X, Y)] \\ Y - \text{proj}_{\mathcal{Y}}[Y + \nabla_Y f(X, Y)] \end{array} \right] \quad (14)$$

where proj denotes the convex projection operator. Clearly, when $\nabla\mathcal{L}(X, Y) = 0$, then a first-order stationary solution of the problem equation 4 is obtained.

**Theorem 1.** *Suppose that the sequence $(X^r, Y^r)$ is generated by equation 10 and $\rho, \beta$ satisfy the conditions equation 12. For a given small constant $\epsilon$, let $T(\epsilon)$ denote the iteration index satisfying the following inequality*

$$T(\epsilon) \triangleq \min\{r \mid \|\nabla\mathcal{L}(X^r, Y^r)\|^2 \leq \epsilon, r \geq 1\}. \quad (15)$$

*Then there exists some constant $C > 0$ such that*

$$\epsilon \leq \frac{C(\mathcal{P}^1 - \underline{\mathcal{P}})}{T(\epsilon)} \quad (16)$$

*where $\underline{\mathcal{P}}$ denotes the lower bound of $\mathcal{P}^r$.*

The above result shows that our proposed algorithm converges to the first-order stationary point of the original problem in a sublinear rate.

## 4 EXPERIMENTS

We conduct several experiments to illustrate our results. The first two examples are on GANs with both synthetic data and MNIST dataset, and the last one (supplemental material) is on multi-task learning with real data. Our intention is by no means to show our algorithm generates superior samples than other methods. Our main goal is to show that our first-order primal-dual algorithm would converge at least to a local solution. All experiments are implemented on a NVIDIA TITAN Xp.

Table 1: Comparison of time elapsed (in seconds) for different dimensions.

|  | 5D | 10D | 20D | 50D |
|---|---|---|---|---|
| ours | 130.27 | 134.76 | 135.47 | 380.98 |
| Sanjabi et al. | 524.00 | 543.52 | 545.18 | 1603.68 |

## 4.1 GAN IN LQG SETTING

In the LQG setting, as discussed in Section 2.2.1, the generator is modeled by a linear map $G(x) = \theta x$ with parameter $\theta$ and the discriminator is parametrized by a positive definite matrix $A$. The seed $\mathbf{x}$ is a zero-mean random variable with unit covariance. We randomly generate a positive definite matrix $\Sigma_{\mathbf{y}}$ as our covariance of the data samples. The solution to this GAN problem satisfies $A = I$, $\theta\theta^T = \Sigma_{\mathbf{y}}$.

We implement our algorithm with different step-sizes. The results are shown in Figure 1 for 20 dimensional data, from which we see that the algorithm converges when the step-size is sufficiently small, while *diverges* for a large step-size. We also compare our algorithm with the one proposed in Sanjabi et al. (2018), which requires solving the inner maximization problem in each iteration. As can be seen from the last plot of Figure 1, while these two algorithms take a similar number of iterations to converge, the total time consumption is more for Sanjabi et al. (2018) as it takes more time to solve the maximization problem than to update the parameter one step along the gradient direction. A detailed comparison is given in Table 1.

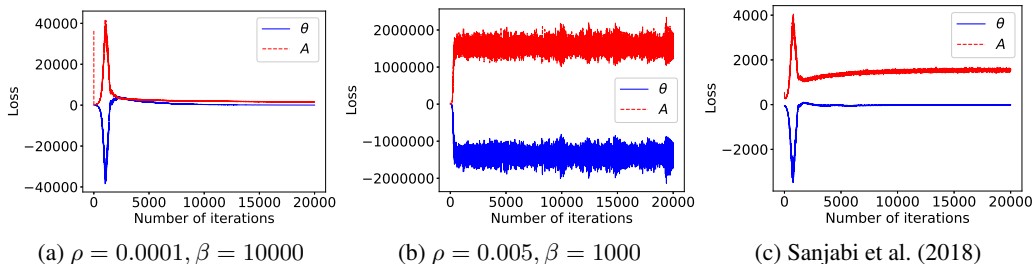

(a) $\rho = 0.0001, \beta = 10000$     (b) $\rho = 0.005, \beta = 1000$     (c) Sanjabi et al. (2018)

Figure 1: (a) converge and (b) diverge show the effect of step-sizes; (c) is based on the algorithm in Sanjabi et al. (2018).

To visualize the effectiveness of LQG GAN, we consider the problem in 2 and 3 dimensional spaces. We plot the samples corresponding to both the learned and the real covariance matrices in Figure 2. Clearly, the learned models match the underlying truth.

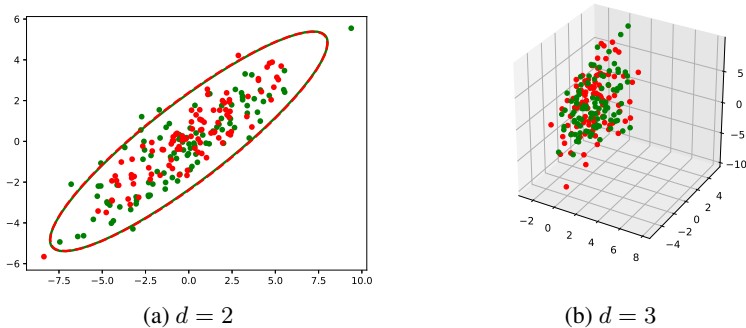

(a) $d = 2$             (b) $d = 3$

Figure 2: Real (green) vs generated (red) samples

### 4.2 GANs with discriminators linear on features

We test the GANs framework with discriminators linear on features (discussed in Section 2.2.2) on the MNIST (LeCun et al., 1998) data of size $28 \times 28$. The network architecture of the generator is same as DCGAN (Radford et al., 2015). To get the reasonable basis functions for the discriminator, we first train a Wasserstein GAN model with a subset of the MNIST data for a small number (5k) of iterations. We then use the last hidden layer of the discriminator as our bases. We implement our algorithm with a different number of basis and the results are shown in Figures 3 and 4. We have two major observations here: i) the GANs with discriminators linear on features generate reasonable samples and the performance improves as we increase the number of bases; ii) the algorithm converges with a small step-size while diverges for a large step-size.

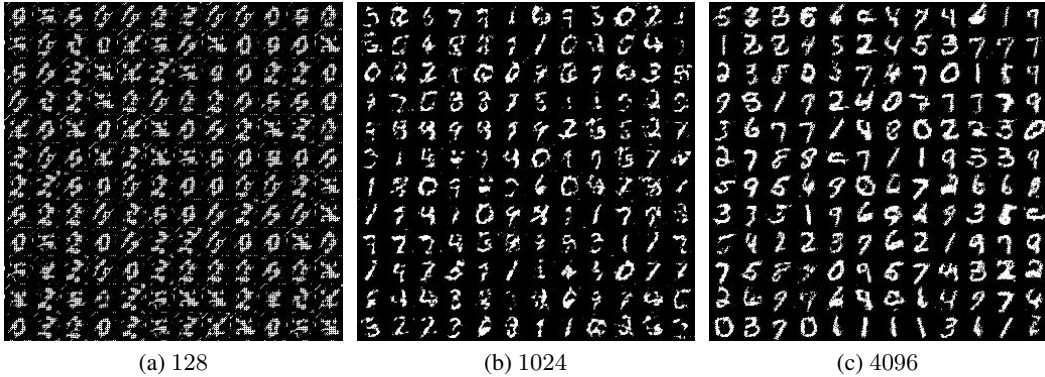

(a) 128      (b) 1024      (c) 4096

Figure 3: Generated samples with different number (a) 128, (b) 1024, and (c) 4096 of basis functions.

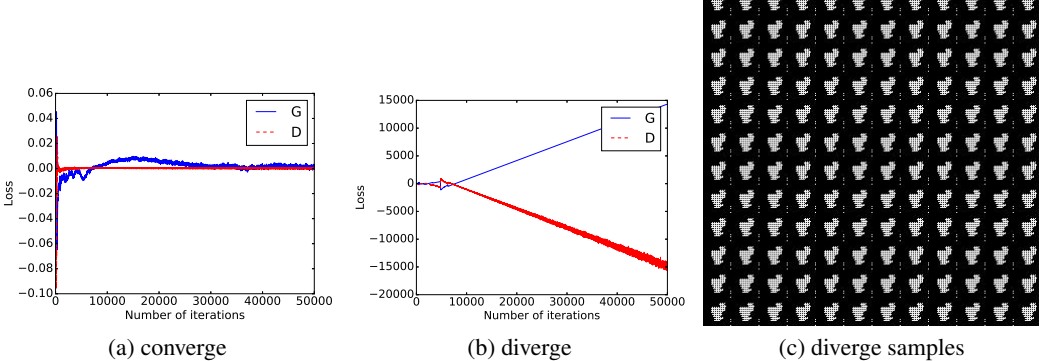

(a) converge      (b) diverge      (c) diverge samples

Figure 4: Effect of stepsizes: the algorithm converge for small stepsize and diverge for large stepsize

## 5 Conclusion

In this work, we presented a convergence result for a first-order algorithm on a class of non-convex max-min optimization problems that arise in many machine learning applications such as generative adversarial networks and multi-task learning. To the best of our knowledge, this is the first convergence result for this type of primal-dual algorithms. Our results allow us to analyze GANs with neural network generator as well as general multi-task non-convex supervised learning problems. A critical assumption we made is that the inner maximization loop is a strictly convex problem. For applications in GANs, our assumptions require the discriminator to be a linear combination of predefined basis functions. Extending this to the most general cases where the discriminator is a neural network requires further investigations and will be a future research topic.

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

# A  ROBUST MULTI-TASK MACHINE LEARNING

Multi-task machine learning (Qian et al., 2018) aims at learning a single model that would work for several different machine learning tasks. Let

$$\min_W \boldsymbol{E}\{f_1(\mathbf{x}^1, W)\}, \cdots, \min_W \boldsymbol{E}\{f_n(\mathbf{x}^n, W)\}$$

be $n$ supervised learning problems, then a multi-task formulation is

$$\min_W \sum_{i=1}^n p_i \boldsymbol{E}\{f_i(\mathbf{x}^i, W)\}, \tag{17}$$

where $p$ is a probability vector to weight the tasks. A common choice is the uniform distribution $\bar{p} = [1/n, \ldots, 1/n]$. To improve the worst case performance, one can change $p$ adaptively and attain the max-min formulation

$$\min_W \max_p \{\sum_{i=1}^n p_i \boldsymbol{E}\{f_i(\mathbf{x}^i, W)\} - \lambda D(p, \bar{p}) + h(W)\}, \tag{18}$$

where $D$ is a distance function to regularize $p$. Here we have also added an regularization term on $W$.

A closely related topic is the distributional robustness (Namkoong & Duchi, 2016) problem

$$\min_W \max_{p \in \mathcal{P}} \sum_{i=1}^n p_i f(x_i, W), \tag{19}$$

where $\mathcal{P}$ is a subset of the space of probability vectors. Relaxing the hard constraint on $p$ points to a regularized version

$$\min_W \max_p \sum_{i=1}^n p_i f(x_i, W) - \lambda D(p, \bar{p}). \tag{20}$$

While $p$ represents weights on different tasks in multi-task learning, it describes data distribution in distributional robustness. It beautifully incorporates data uncertainties in the learning problems. This formulation has also applications in adversarial learning Madry et al. (2018), where $p_i f(x_i, W)$ denotes the loss given by data $x_i$ after adversarial reweighing through the inner maximization problem. The goal of the outer minimization is to learn a model with parameters $W$ such that the worst case loss is minimized.

## A.1  NUMERICAL EXAMPLE

We consider two supervised learning tasks with MNIST (LeCun et al., 1998) dataset and CIFAR10 (Krizhevsky & Hinton, 2009) data set. We seek a single neural network that works for these two completely unrelated problems (see Section A). First, we convert the MNIST data from $28 \times 28$ gray images to $32 \times 32$ color images so that it is in the same format as CIFAR10. We use a standard AlexNet (Krizhevsky et al., 2012) as our model. We train the model with the robust multi-task learning framework and compare it to the results from three other methods: train with MNIST only, train with CIFAR10 only and train with both data sets but with even weight $[0.5, 0.5]$. The batch size we use is 128.

The results are presented in Table 2. The last row is the results for the robust multi-task learning, and the "even mixture" in the second last row standards for uniform weight $[0.5, 0.5]$ between the two tasks. We see that the result of using robust multi-task learning framework is better than the one with even weight. Moreover, its performance on each task is comparable to that trained from a single data set alone. The optimal $p$ is $[0.205, 0.795]$, where the first value is for the MNIST dataset. We also implement our algorithm with different levels $\lambda$ of regularizations. The results are shown in Figure 5, where we display the loss functions in the first two plots and the weight for MNIST in the last plot. Even though changing $\lambda$ doesn't affect the convergence rate that much, it changes the optimal $p$ significantly.

Table 2: Multi-task Learning: Comparison of Accuracies

| Method | MNIST | | CIFAR10 | |
|--------|----------|--------|----------|--------|
| | Training | Test | Training | Test |
| MNIST | 99.98 | 99.33 | 9.24 | 9.16 |
| CIFAR10 | 10.25 | 10.10 | 99.75 | 76.59 |
| Even Mixture | 99.96 | 99.40 | 99.42 | 74.97 |
| OPT Mixture | 99.98 | **99.35** | 99.85 | **76.29** |

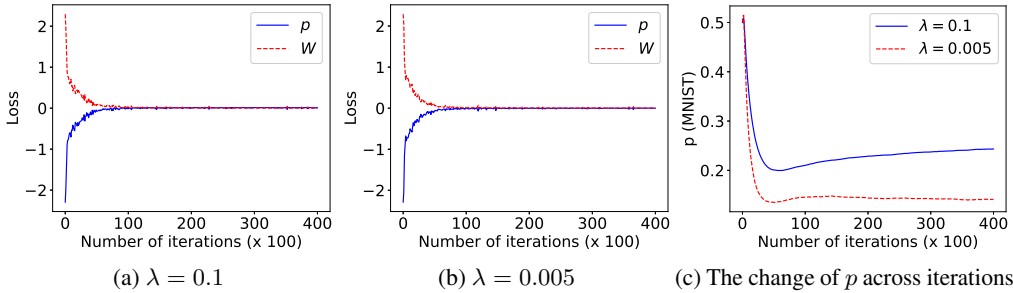

(a) $\lambda = 0.1$      (b) $\lambda = 0.005$      (c) The change of $p$ across iterations

Figure 5: Effect of regularization. Here $p$ is the weight, $W$ is the parameter and $\lambda$ is the regularization constant. The regularizer $D$ is taken to be the Kullback-Leibler divergence.

## B    CONVERGENCE ANALYSIS OF THE PRIMAL DUAL ALGORITHM

In our convergence analysis of the primal-dual algorithm, we use the optimality conditions of $X^r$- and $Y^r$-subproblems repeatedly so that the quantities of measuring the size of the difference of the iterates, e.g., $\|X^{r+1} - X^r\|$ and $\|Y^{r+1} - Y^r\|$, can be obtained. Also, for the simplicity of the notation, we have the following definitions:

$$h(X) \triangleq \frac{1}{n}\sum_{i=1}^{n} h_i(X), \quad g(X) \triangleq \frac{1}{n}\sum_{i=1}^{n} g_i(X), \quad l(Y) \triangleq \frac{1}{n}\sum_{i=1}^{n} l_i(Y). \tag{21}$$

Then, we give the optimality condition of $X^r$-subproblem and $Y^r$-subproblem as follows,

$$\left\langle \nabla h(X^r) + \sum_{j,k} \nabla_X g_{j,k}(X^r) Y_{j,k}^r + \beta(X^{r+1} - X^r), X^{r+1} - X \right\rangle \leq 0, \forall X \in \mathcal{X}, \tag{22}$$

$$Y^{r+1} = Y^r + \rho\epsilon^{r+1} + \rho\left(g(X^{r+1}) - \nabla_Y l(Y^r)\right). \tag{23}$$

where $-\epsilon^{r+1}$ denotes the subgradient of the convex indicator function $\mathbf{1}(y^{r+1} \in \mathcal{Y})$, $Y_{j,k}^r$ denotes the entry at the $j$ row and $k$th column of $Y^r$, and $g_{j,k}(X^r)$ denotes the matrix value function mapping from $X^r$ to the value at the $j$th row and $k$th column of $g(X^r)$. Note that equation 23 is also equivalent to

$$\left\langle -\nabla_Y l(Y^r) + g(X^{r+1}) - \frac{1}{\rho}(Y^{r+1} - Y^r), Y^{r+1} - Y \right\rangle \geq 0, \; \forall Y \in \mathcal{Y}. \tag{24}$$

Before going to the details, we will first introduce the following lemma that characterizes the upper bound of $\|Y^r\|$.

**Lemma 3.** *Let $(X^r, Y^r)$ be a sequence generated by equation 10. The size of $Y^r$ is upper bounded by some constant number denoted by $\sigma_Y$.*

*Proof.* Let $\zeta(X,Y) \triangleq l(Y) - \langle g(X), Y \rangle$. Define:

$$Y_1^* \triangleq \arg\min_{Y \in \mathcal{Y}} \zeta(X_1, Y) \quad Y_2^* \triangleq \arg\min_{Y \in \mathcal{Y}} \zeta(X_2, Y) \tag{25}$$

From the optimality conditions, we know that

$$\langle \nabla_Y \zeta(X_1, Y_1^*), Y_2^* - Y_1^* \rangle \geq 0 \quad \langle \nabla_Y \zeta(X_2, Y_2^*), Y_1^* - Y_2^* \rangle \geq 0. \tag{26}$$

Adding these two inequalities, we have

$$\langle \nabla_Y \zeta(X_1, Y_1^*) - \nabla_Y \zeta(X_2, Y_2^*), Y_2^* - Y_1^* \rangle \geq 0, \tag{27}$$

which implies that

$$
\begin{aligned}
&L_{g,1}\|X_1 - X_2\|\|Y_1^* - Y_2^*\| \\
&\geq \langle g(X_2) - g(X_1), Y_1^* - Y_2^* \rangle \\
&= \langle \nabla_Y \zeta(X_2, Y_2^*) - \nabla_Y \zeta(X_1, Y_2^*), Y_1^* - Y_2^* \rangle \\
&\geq \langle \nabla_Y \zeta(X_1, Y_1^*) - \nabla_Y \zeta(X_1, Y_2^*), Y_1^* - Y_2^* \rangle \\
&= \langle \nabla l(Y_1^*) - \nabla l(Y_2^*), Y_1^* - Y_2^* \rangle \\
&\geq \gamma \|Y_1^* - Y_2^*\|^2
\end{aligned}
$$

where in the first inequality we used the Lipschitz continuity; in the last inequality we used the strong convexity of function $l(Y)$. Therefore, we have

$$\|Y_1^* - Y_2^*\| \leq \frac{L_{g,1}}{\gamma} \|X_1 - X_2\|. \tag{28}$$

Since $X \in \mathcal{X}$, with $\|X\|$ bounded by $\sigma_X$, we have the claim that the distance between any two $Y_1^*$ and $Y_2^*$ are bounded by $2L_{g,1}\sigma_X/\gamma$. In other words, $Y_{X^r}^*, \forall X^r \in \mathcal{X}$ are within a compact sets, where $Y_{X^r}^* \triangleq \arg\min_{Y \in \mathcal{Y}} \zeta(X^r, Y)$ and the radius of the set is upper bounded by $2L_{g,1}\sigma_X/\gamma$.

Let $Y^{r+1}$ denote the $r+1$th iterate of the projected gradient descent method of solving the dual problem which is parameterized by $X^{r+1}$. Because the dual problem is strongly convex, it is standard to show that

$$\|Y^{r+1} - Y_{X^{r+1}}^*\| \leq \|Y^r - Y_{X^{r+1}}^*\|. \tag{29}$$

This implies that the distance between the iterate and the set is not increasing. The proof is complete. □

Throughout our analysis, we used $\sigma_Y \geq \|Y^r\|_1, \forall r$ where $\|Y\|_1 \triangleq \sum_{i,j} |Y_{i,j}|$. Note that $\|Y\|_1 \geq \|Y\|_F$. Based on this definition and the previous lemma, it is easy to check that the following holds

$$\|\nabla f(X, Y) - \nabla f(Z, Y)\| \leq (L_X + \sigma_Y L_{g,2})\|X - Z\| \tag{30}$$

where $L_X$ and $L_{g,2}$ are two constants defined after equation 4.

## B.1    PROOF OF LEMMA 1

### B.1.1    PRIMAL PROBLEM

Define

$$f(X, Y) \triangleq h(X) - l(Y) + \langle g(X), Y \rangle. \tag{31}$$

First, suppose that we choose $\beta$ large enough such that $\beta \geq L_X + \sigma_Y L_{g,2}$. Then we have the following estimate of the descent of the objective value:

$$
\begin{aligned}
&f(X^{r+1}, Y^r) - f(X^r, Y^r) \\
=&h(X^{r+1}) + \langle g(X^{r+1}), Y^r \rangle - h(X^r) - \langle g(X^r), Y^r \rangle \\
\overset{(a)}{\leq}& \langle \nabla_X h(X^r) + \sum_{i,j} \nabla_X g_{i,j}(X^r) Y_{j,k}^r, X^{r+1} - X^r \rangle + \frac{L_X + \sigma_Y L_{g,2}}{2} \|X^{r+1} - X^r\|^2 \\
\overset{(b)}{\leq}& -\frac{\beta}{2}\|X^{r+1} - X^r\|^2
\end{aligned} \tag{32}
$$

when in $(a)$ we used the equation 30; In $(b)$ wej used the optimality condition equation 22, and we choose $\beta \geq L_X + \sigma_Y L_{g,2}$.

### B.1.2 DUAL PROBLEM

For the dual problem, we have

$$
\begin{aligned}
&f(X^{r+1}, Y^{r+1}) - f(X^{r+1}, Y^r) \\
&= -l(Y^{r+1}) + \langle g(X^{r+1}), Y^{r+1} - Y^r \rangle + l(Y^r) - (\mathbf{1}(Y^{r+1} \in \mathcal{Y}) - \mathbf{1}(Y^r \in \mathcal{Y})) \\
&\overset{(a)}{\leq} -\langle \nabla_Y l(Y^r) - \epsilon^r, Y^{r+1} - Y^r \rangle + \langle g(X^{r+1}), Y^{r+1} - Y^r \rangle \\
&= -\langle \nabla_Y l(Y^r) - \epsilon^{r+1}, Y^{r+1} - Y^r \rangle - \langle \epsilon^{r+1} - \epsilon^r, Y^{r+1} - Y^r \rangle + \langle g(X^{r+1}), Y^{r+1} - Y^r \rangle \\
&\overset{(b)}{=} \frac{1}{\rho} \langle Y^{r+1} - Y^r, Y^{r+1} - Y^r \rangle - \langle \epsilon^{r+1} - \epsilon^r, Y^{r+1} - Y^r \rangle \\
&\overset{(c)}{\leq} \frac{1}{\rho} \langle Y^{r+1} - Y^r, Y^{r+1} - Y^r \rangle + \frac{1}{2} \left( \left( L_Y + L_{g,1} - \frac{1}{\rho} \right) \|Y^{r+1} - Y^r\|^2 \right. \\
&\quad + \frac{L_{g,1}}{2} \|X^{r+1} - X^r\|^2 + \frac{1}{2} \left( L_Y + \frac{1}{\rho} \right) \|Y^r - Y^{r-1}\|^2 \right) \\
&= \frac{1}{2} \left( \left( L_Y + L_{g,1} + \frac{1}{\rho} \right) \|Y^{r+1} - Y^r\|^2 + \frac{L_{g,1}}{2} \|X^{r+1} - X^r\|^2 \right. \\
&\quad + \frac{1}{2} \left( L_Y + \frac{1}{\rho} \right) \|Y^r - Y^{r-1}\|^2
\end{aligned}
$$

where in $(a)$ we used the convexity of both function $l(X)$ and indicator function, i.e., $l(Y^{r+1}) + \mathbf{1}(Y^{r+1} \in \mathcal{Y}) - (l(Y^r) + \mathbf{1}(Y^r \in \mathcal{Y})) \geq \langle \nabla_Y l(Y^r) - \epsilon^r, Y^{r+1} - Y^r \rangle$, in $(b)$ we used equation 23 and $(c)$ is true because the following relations:

1. First, from equation 23 we have

$$
\begin{aligned}
Y^{r+1} &= Y^r + \rho(g(X^{r+1}) - \nabla l(Y^r) + \epsilon^{r+1}), &(33) \\
Y^r &= Y^{r-1} + \rho(g(X^r) - \nabla l(Y^{r-1}) + \epsilon^r), &(34)
\end{aligned}
$$

which imply

$$
\epsilon^{r+1} - \epsilon^r = \frac{1}{\rho}(Y^{r+1} - Y^r - (Y^r - Y^{r-1})) - \big(g(X^{r+1}) - g(X^r)\big) + \nabla l(Y^r) - \nabla l(Y^{r-1}). \tag{35}
$$

2. Second, we have

$$
\begin{aligned}
&\langle \epsilon^{r+1} - \epsilon^r, Y^{r+1} - Y^r \rangle \\
&= \frac{1}{\rho} \langle V^{r+1}, Y^{r+1} - Y^r \rangle - \langle g(X^{r+1}) - g(X^r), Y^{r+1} - Y^r \rangle \\
&\quad - \langle \nabla_Y l(Y^r) - \nabla_Y l(Y^{r-1}), Y^{r+1} - Y^r \rangle \\
&\overset{(a)}{\geq} -\frac{1}{2\rho}(\|Y^r - Y^{r-1}\|^2 - \|Y^{r+1} - Y^r\|^2) - \frac{1}{2}(L_Y + L_{g,1})\|Y^{r+1} - Y^r\|^2 \\
&\quad - \frac{1}{2}L_{g,1}\|X^{r+1} - X^r\|^2 - \frac{1}{2}L_Y\|Y^r - Y^{r-1}\|^2 \\
&\geq -\frac{1}{2}\left( \left( L_Y + L_{g,1} - \frac{1}{\rho} \right) \|Y^{r+1} - Y^r\|^2 - \frac{L_{g,1}}{2}\|X^{r+1} - X^r\|^2 \right. \\
&\quad - \frac{1}{2}\left( L_Y + \frac{1}{\rho} \right) \|Y^r - Y^{r-1}\|^2 \right)
\end{aligned}
$$

where in $(a)$ we used Cauchy–Schwarz inequality.

Finally, combing the above results, we have

$$f(X^{r+1}, Y^{r+1}) - f(X^r, Y^r)$$

$$\leq \underbrace{-\frac{\beta}{2}\|X^{r+1} - X^r\|^2}_{\text{primal descent}} + \frac{1}{2}\left(L_Y + L_{g,1} + \frac{1}{\rho}\right)\|Y^{r+1} - Y^r\|^2$$

$$+ \frac{L_{g,1}}{2}\|X^{r+1} - X^r\|^2 + \frac{1}{2}\left(L_Y + \frac{1}{\rho}\right)\|Y^r - Y^{r-1}\|^2. \tag{36}$$

## B.2 PROOF OF LEMMA 2

*Proof.* Let $W^{r+1} \triangleq (X^{r+1} - X^r) - (X^r - X^{r-1})$. Subtracting equation 22 at iteration $r-1$ from itself at the $r$th iteration, we have the successive difference of equation 23, i.e.,

$$\Big\langle \nabla_X h(X^r) - \nabla_X h(X^{r-1})$$

$$+ \underbrace{\left(\sum_{j,k}\nabla g_{j,k}(X^r)Y_{j,k}^r - \sum_{j,k}\nabla g_{j,k}(X^{r-1})Y_{j,k}^{r-1}\right)}_{\triangleq \mathsf{A}^r} + \beta W^{r+1}, X^{r+1} - X^r\Big\rangle \leq 0. \tag{37}$$

Similarly, from equation 23, we also have

$$\underbrace{Y^{r+1} - Y^r - (Y^r - Y^{r-1})}_{\triangleq V^{r+1}} = \rho\left(\epsilon^{r+1} - \epsilon^r\right)$$

$$+ \rho\left(g(X^{r+1}) - g(X^r)\right) - \rho\left(\nabla l(Y^r) - \nabla l(Y^{r-1})\right). \tag{38}$$

### B.2.1 INDUCTION OF THE SIZE OF THE SUCCESSIVE ITERATES

First, we note the following key inequality

$$\langle W^{r+1}, X^{r+1} - X^r\rangle = \frac{1}{2}\|X^{r+1} - X^r\|^2 - \frac{1}{2}\|X^r - X^{r-1}\|^2 + \frac{1}{2}\|W^{r+1}\|^2. \tag{39}$$

Second, we can get the lower bound of $\langle \mathsf{A}^r, X^{r+1} - X^r\rangle$ as follows:

$$\Big\langle \sum_{j,k}\nabla_X g_{j,k}(X^r)Y_{j,k}^r - \sum_{j,k}\nabla g_{j,k}(X^{r-1})Y_{j,k}^{r-1}, X^{r+1} - X^r\Big\rangle \tag{40}$$

$$= \Big\langle \sum_{j,k}\nabla g_{j,k}(X^r)\left(Y_{j,k}^r - Y_{j,k}^{r-1}\right), X^{r+1} - X^r\Big\rangle$$

$$+ \Big\langle \sum_{j,k}\left(\nabla g_{j,k}(X^r) - \nabla g_{j,k}(X^{r-1})\right)Y_{j,k}^{r-1}, X^{r+1} - X^r\Big\rangle \tag{41}$$

$$= \sum_{m,n}\sum_{j,k}\nabla_{X_{m,n}} g_{j,k}(X^r)\left(Y_{j,k}^r - Y_{j,k}^{r-1}\right)(X_{m,n}^{r+1} - X_{m,n}^r)$$

$$+ \sum_{m,n}\sum_{j,k}\left(\nabla_{X_{m,n}} g_{j,k}(X^r) - \nabla_{X_{m,n}} g_{j,k}(X^{r-1})\right)Y_{j,k}^{r-1}(X_{m,n}^{r+1} - X_{m,n}^r) \tag{42}$$

$$= \underbrace{\sum_{j,k}\langle \nabla_X g_{j,k}(X^r), X^{r+1} - X^r\rangle\left(Y_{j,k}^r - Y_{j,k}^{r-1}\right)}_{\triangleq \mathsf{A}_1^r}$$

$$+ \underbrace{\sum_{j,k}\langle \nabla_X g_{j,k}(X^r) - \nabla_X g_{j,k}(X^{r-1}), X^{r+1} - X^r\rangle Y_{j,k}^{r-1}}_{\triangleq \mathsf{A}_2^r}. \tag{43}$$

In the above derivation, we basically changed the order of summations.

From the mean value theorem, we know that there exits $\widetilde{X}^{r+1}_{(m,n)} \in [X^r, X^{r+1}]$ such that

$$g_{m,n}(X^{r+1}) - g_{m,n}(X^r) = \left\langle \nabla_X g_{m,n}(\widetilde{X}^{r+1}_{(m,n)}), X^{r+1} - X^r \right\rangle. \tag{44}$$

Hence, $\mathsf{A}^r_1$ becomes

$$\sum_{j,k} \langle \nabla_X g_{j,k}(X^r), X^{r+1} - X^r \rangle (Y^r_{j,k} - Y^{r-1}_{j,k})$$

$$= \sum_{j,k} \langle \nabla_X g_{j,k}(X^r) - \nabla_X g_{j,k}(\widetilde{X}^{r+1}), X^{r+1} - X^r \rangle (Y^r_{j,k} - Y^{r-1}_{j,k}) \tag{45}$$

$$+ \sum_{j,k} \langle \nabla_X g_{j,k}(\widetilde{X}^{r+1}), X^{r+1} - X^r \rangle (Y^r_{j,k} - Y^{r-1}_{j,k})$$

$$= \sum_{j,k} \langle \nabla_X g_{j,k}(X^r) - \nabla_X g_{j,k}(\widetilde{X}^{r+1}), X^{r+1} - X^r \rangle (Y^r_{j,k} - Y^{r-1}_{j,k})$$

$$+ \sum_{j,k} (g_{j,k}(X^{r+1}) - g_{j,k}(X^r))(Y^r_{j,k} - Y^{r-1}_{j,k})$$

$$\geq - \|Y^r - Y^{r-1}\|_1 \|\widetilde{X}^{r+1} - X^r\| \|X^{r+1} - X^r\| L_{g,2}$$

$$+ \left\langle \frac{V^{r+1}}{\rho} + \nabla_Y l(Y^r) - \nabla_Y l(Y^{r-1}), Y^r - Y^{r-1} \right\rangle - \langle \epsilon^{r+1} - \epsilon^r, Y^r - Y^{r-1} \rangle$$

$$\overset{(a)}{\geq} - \|Y^r - Y^{r-1}\|_1 \|X^{r+1} - X^r\|^2 L_{g,2} + \left\langle \frac{V^{r+1}}{\rho} + \nabla_Y l(Y^r) - \nabla_Y l(Y^{r-1}), Y^r - Y^{r-1} \right\rangle$$

$$- \langle \epsilon^{r+1} - \epsilon^r, Y^r - Y^{r-1} \rangle$$

where in $(a)$ we used Cauchy–Schwarz inequality.

Combining $\mathsf{A}^r_1$ and $\mathsf{A}^r_2$, we can get the lower bound of $\langle \mathsf{A}^r, X^{r+1} - X^r \rangle$, i.e.,

$$\langle \mathsf{A}^r, X^{r+1} - X^r \rangle$$

$$\geq - \|Y^r - Y^{r-1}\|_1 \|X^{r+1} - X^r\|^2 L_{g,2} + \left\langle \frac{V^{r+1}}{\rho} + \nabla_Y l(Y^r) - \nabla_Y l(Y^{r-1}), Y^r - Y^{r-1} \right\rangle$$

$$- \langle \epsilon^{r+1} - \epsilon^r, Y^r - Y^{r-1} \rangle + \sum_{j,k} \langle \nabla_X g_{j,k}(X^r)) - \nabla_X g_{j,k}(X^{r-1})), X^{r+1} - X^r \rangle Y^{r-1}_{j,k}$$

$$\geq - \|Y^r - Y^{r-1}\|_1 \|X^{r+1} - X^r\|^2 L_{g,2} + \left\langle \frac{V^{r+1}}{\rho} - \epsilon^{r+1} - \epsilon^r, Y^r - Y^{r-1} \right\rangle$$

$$+ \langle \nabla_Y l(Y^r) - \nabla_Y l(Y^{r-1}), Y^r - Y^{r-1} \rangle$$

$$- \frac{1}{2} \|Y^{r-1}\|_1 L_X \left( \|X^r - X^{r-1}\|^2 + \|X^{r+1} - X^r\|^2 \right).$$

To further bound the above terms, we will give the upper bound of $\langle Y^r - Y^{r-1}, V^{r+1}/\rho - (\epsilon^{r+1} - \epsilon^r) \rangle$ and $\langle \nabla_Y l(Y^r) - \nabla_Y l(Y^{r-1}), Y^r - Y^{r-1} \rangle$ separately in the following two steps.

**Step 1).** First, we have

$$\left\langle Y^r - Y^{r-1}, \frac{V^{r+1}}{\rho} - (\epsilon^{r+1} - \epsilon^r) \right\rangle \tag{46}$$

$$= \left\langle Y^r - Y^{r-1} - (Y^{r+1} - Y^r) + Y^{r+1} - Y^r, \frac{V^{r+1}}{\rho} - (\epsilon^{r+1} - \epsilon^r) \right\rangle \tag{47}$$

$$= \rho \left\langle \frac{V^{r+1}}{\rho}, \frac{V^{r+1}}{\rho} - (\epsilon^{r+1} - \epsilon^r) \right\rangle - \left\langle Y^{r+1} - Y^r, \frac{V^{r+1}}{\rho} - (\epsilon^{r+1} - \epsilon^r) \right\rangle$$

$$\overset{(a)}{=} \frac{\rho}{2} \left( \left\| \frac{V^{r+1}}{\rho} \right\|^2 + \left\| \frac{V^{r+1}}{\rho} - (\epsilon^{r+1} - \epsilon^r) \right\|^2 - \|\epsilon^{r+1} - \epsilon^r\|^2 \right)$$

$$- \frac{1}{2\rho} \left( \|Y^{r+1} - Y^r\|^2 - \|Y^r - Y^{r-1}\|^2 + \|V^{r+1}\|^2 \right) + \underbrace{\langle Y^{r+1} - Y^r, \epsilon^{r+1} - \epsilon^r \rangle}_{\leq 0} \tag{48}$$

$$\leq \frac{\rho}{2} \left\| \frac{V^{r+1}}{\rho} - (\epsilon^{r+1} - \epsilon^r) \right\|^2 - \frac{1}{2\rho} \|Y^{r+1} - Y^r\|^2 + \frac{1}{2\rho} \|Y^r - Y^{r-1}\|^2 \tag{49}$$

where in $(a)$ we used

$$\langle -\epsilon^{r+1} - (-\epsilon^r), Y^{r+1} - Y^r \rangle \geq 0 \tag{50}$$

due to the fact that $-\epsilon^{r+1}$ is the subgradient of the convex indicator function $\mathbf{1}(y^{r+1} \in \mathcal{Y})$, and also the equality

$$\langle Y^r - Y^{r-1}, V^{r+1} \rangle = \frac{1}{2} \|Y^{r+1} - Y^r\|^2 - \frac{1}{2} \|Y^r - Y^{r-1}\|^2 - \frac{1}{2} \|V^{r+1}\|^2. \tag{51}$$

Further, from the optimality condition of the $Y^r$ subprolbem shown in equation 23, we know

$$\frac{V^{r+1}}{\rho} - (\epsilon^{r+1} - \epsilon^r) = g(X^{r+1}) - g(X^r) - (\nabla_Y l(Y^r) - \nabla_Y l(Y^{r-1})), \tag{52}$$

which gives

$$\frac{\rho}{2} \left\| \frac{V^{r+1}}{\rho} - (\epsilon^{r+1} - \epsilon^r) \right\|^2 \leq \rho L_{g,1}^2 \|X^{r+1} - X^r\|^2 + \rho L_Y^2 \|Y^r - Y^{r-1}\|^2 \tag{53}$$

where we used the Lipschitz continuity.

**Step 2).** Next, we need to quantify the lower bound of $\langle \nabla_Y l(Y^r) - \nabla_Y l(Y^{r-1}), Y^r - Y^{r-1} \rangle$. Since $l(Y)$ is strongly convex, we have

$$\langle \nabla_Y l(Y^r) - \nabla_Y l(Y^{r-1}), Y^r - Y^{r-1} \rangle \geq \gamma \|Y^r - Y^{r-1}\|^2. \tag{54}$$

Combining step 1 and step 2 and equation 46, we have the lower bound of $\langle \mathsf{A}^r, X^{r+1} - X^r \rangle$

$$\langle \mathsf{A}^r, X^{r+1} - X^r \rangle \geq -\|Y^r - Y^{r-1}\|_1 L_{g,2} \|X^{r+1} - X^r\|^2 - \frac{1}{2} \|Y^{r-1}\|_1 L_X \|X^{r+1} - X^r\|^2$$

$$- \frac{1}{2} \|Y^{r-1}\| L_X \|X^r - X^{r-1}\|^2 + \frac{1}{2\rho} \|Y^{r+1} - Y^r\|^2 - \frac{1}{2\rho} \|Y^r - Y^{r-1}\|^2$$

$$- \rho L_{g,1}^2 \|X^{r+1} - X^r\|^2 - \rho L_Y^2 \|Y^{r+1} - Y^r\|^2 + \gamma \|Y^r - Y^{r-1}\|^2. \tag{55}$$

Therefore, dividing equation 37 by $\beta$ and combining equation 39, equation 55, we have

$$
\frac{1}{2}\|X^{r+1} - X^r\|^2 + \frac{1}{2\rho\beta}\|Y^{r+1} - Y^r\|^2 \tag{56}
$$

$$
\leq \frac{1}{2}\|X^r - X^{r-1}\|^2 + \frac{1}{2\rho\beta}\|Y^r - Y^{r-1}\|^2 - \frac{1}{2}\|W^{r+1}\|^2
$$

$$
+ \frac{2L_{g,2}\|Y^r - Y^{r-1}\|_1 + L_X\|Y^{r-1}\|_1 + L_X + 2\rho L_{g,1}^2}{2\beta}\|X^{r+1} - X^r\|^2
$$

$$
+ \frac{L_X\|Y^{r-1}\|_1 + L_X}{2\beta}\|X^r - X^{r-1}\|^2 - \frac{\gamma - \rho L_Y^2}{\beta}\|Y^r - Y^{r-1}\|^2
$$

$$
\overset{(a)}{\leq} \frac{1}{2}\|X^r - X^{r-1}\|^2 + \frac{1}{2\rho\beta}\|Y^r - Y^{r-1}\|^2 \underbrace{- \frac{\gamma - \rho L_Y^2}{\beta}\|Y^r - Y^{r-1}\|^2}_{\text{dual descent}}
$$

$$
+ \frac{5\sigma_Y L_X + L_{g,2} + 2\rho L_{g,1}^2}{2\beta}\|X^{r+1} - X^r\|^2 + \frac{L_X(\sigma_Y + 1)}{2\beta}\|X^r - X^{r-1}\|^2 \tag{57}
$$

where in $(a)$ we used $\sigma_Y \geq \|Y^r\|_1, \forall r$.

### B.2.2 Potential Function

Rearranging equation 57, we have

$$
\mathcal{Q}^{r+1} \triangleq \left(\frac{L_X(\sigma_Y + 1)}{2\beta} + \frac{1}{2}\right)\|X^{r+1} - X^r\|^2 + \left(\frac{1}{2\rho\beta} - \frac{\gamma - \rho L_Y^2}{2\beta}\right)\|Y^{r+1} - Y^r\|^2
$$

$$
\leq \left(\frac{L_X(\sigma_Y + 1)}{2\beta} + \frac{1}{2}\right)\|X^r - X^{r-1}\|^2 + \left(\frac{1}{2\rho\beta} - \frac{\gamma - \rho L_Y^2}{2\beta}\right)\|Y^r - Y^{r-1}\|^2
$$

$$
- \frac{\gamma - \rho L_Y^2}{2\beta}\|Y^{r+1} - Y^r\|^2 + \frac{(6\sigma_Y + 1)L_X + L_{g,1} + 2\rho L_{g,1}^2}{2\beta}\|X^{r+1} - X^r\|^2
$$

$$
= \mathcal{Q}^r + \frac{(6\sigma_Y + 1)L_X + L_{g,2} + 2\rho L_{g,1}^2}{2\beta}\|X^{r+1} - X^r\|^2
$$

$$
- \frac{\gamma - \rho L_Y^2}{2\beta}\|Y^{r+1} - Y^r\|^2 - \frac{\gamma - \rho L_Y^2}{2\beta}\|Y^r - Y^{r-1}\|^2. \tag{58}
$$

Define a potential function $\mathcal{P}^{r+1} \triangleq (X^{r+1}, Y^{r+1}) + \underline{d}\mathcal{Q}^{r+1}$. We can obtain

$$
\mathcal{P}^{r+1} - \mathcal{P}^r \leq - \left(\underbrace{\frac{\beta}{2} - \frac{L_{g,1}}{2} - \frac{\underline{d}((6\sigma_Y + 1)L_X + L_{g,2} + 2\rho L_{g,1}^2)}{2\beta}}_{\triangleq c_1}\right)\|X^{r+1} - X^r\|^2
$$

$$
- \left(\underbrace{\frac{\underline{d}(\gamma - \rho L_Y^2)}{2\beta} - \frac{L_Y + L_{g,1} + \frac{1}{\rho}}{2}}_{\triangleq c_2}\right)\|Y^{r+1} - Y^r\|^2
$$

$$
- \left(\underbrace{\frac{\underline{d}(\gamma - \rho L_Y^2)}{2\beta} - \frac{L_Y + \frac{1}{\rho}}{2}}_{\triangleq c_3}\right)\|Y^r - Y^{r-1}\|^2. \tag{59}
$$

### B.2.3 CONVERGENCE CONDITIONS

If the following conditions hold, i.e., $c_1, c_2, c_3 > 0$, then the potential function has a sufficient decrease at each iteration.

$$\frac{\beta}{2} - \frac{L_{g,1}}{2} - \frac{\underline{d}(6\sigma_Y + 1)L_X + L_{g,2} + 2\rho L_{g,1}^2)}{2\beta} > 0, \tag{60}$$

$$\frac{\underline{d}}{\beta}(\gamma - \rho L_Y^2) - (L_Y + L_{g,1} + \frac{1}{\rho}) > 0. \tag{61}$$

To show that $\exists \underline{d} \geq 0$ such that

$$\begin{bmatrix} \beta - L_{g,1} & 0 \\ 0 & L_Y + L_{g,1} + \frac{1}{\rho} \end{bmatrix} + \underline{d} \begin{bmatrix} -\frac{1}{\beta}\left((6\sigma_Y + 1)L_X + L_{g,2} + 2\rho L_{g,1}^2\right) & 0 \\ 0 & \frac{1}{\beta}(\gamma - \rho L_Y^2) \end{bmatrix} \succ 0$$

by $\mathcal{S}$-procedure it is sufficiently to show the set $\widetilde{\mathcal{X}} = \{x | (\beta - L_{g,1})x^2 < L_Y + L_{g,1} + \frac{1}{\rho}, ((6\sigma_Y + 1)L_X + L_{g,2} + 2\rho L_{g,1}^2)x^2 > (\gamma - \rho L_Y^2)\}$ is empty.

Hence, we require that

1.

$$\gamma - \rho L_Y^2 > 0, \tag{62}$$

   which gives

$$\rho < \frac{\gamma}{L_Y^2}; \tag{63}$$

2. $\beta \geq L_{g,1}$;

3. and we also need

$$\beta \geq L_{g,1} + \frac{(L_Y + L_{g,1} + \frac{1}{\rho})((6\sigma_Y + 1)L_X + L_{g,2} + 2\rho L_{g,1}^2)}{\gamma - \rho L_y^2}. \tag{64}$$

Note that we need $\frac{1}{\rho\beta} - \frac{\gamma - \rho L_Y^2}{2\beta} \geq 0$, since the potential function should be greater than 0, meaning that

$$L_Y^2 \rho^2 - \gamma \rho + 2 \geq 0, \tag{65}$$

which is actually satisfied automatically by checking the two roots of the equation $L_Y^2 \rho^2 - \gamma \rho + 2$ (note that $L_Y \geq \gamma$).

Combing the descent lemma shown in Lemma 1, we have the conditions of $\rho, \beta$ shown in Lemma 2 so that the potential function can have a sufficient descent at each iteration. □

### B.3 PROOF OF THEOREM 1

*Proof.* First, we can get the upper bound of $\|\nabla \mathcal{L}(X^r, Y^r)\|$, which is

$$\|\nabla \mathcal{L}(X^r, Y^r)\|$$
$$\leq \|X^{r+1} - X^r\| + \|X^{r+1} - \text{proj}_{\mathcal{X}}(X^r - \nabla_X f(X^r, Y^r))\|$$
$$+ \|Y^{r+1} - Y^r\| + \|Y^{r+1} - \text{proj}_{\mathcal{Y}}(Y^r + \nabla_Y f(X^r, Y^r))\|$$
$$\overset{(a)}{\leq} \|X^{r+1} - X^r\| + \|\text{proj}_{\mathcal{X}}(X^{r+1} - \nabla_X f(X^{r+1}, Y^r)) - \text{proj}_{\mathcal{X}}(X^r - \nabla_X f(X^r, Y^r))\|$$
$$+ \|Y^{r+1} - Y^r\| + \|\text{proj}_{\mathcal{Y}}(Y^{r+1} + \nabla_Y f(X^{r+1}, Y^{r+1})) - \text{proj}_{\mathcal{Y}}(Y^r + \nabla_Y f(X^r, Y^r))\|$$
$$\overset{(b)}{\leq} \|X^{r+1} - X^r\| + \|\nabla_X f(X^{r+1}, Y^r)) - \nabla_X f(X^r, Y^r))\|$$
$$+ \|Y^{r+1} - Y^r\| + \|\nabla_Y f(X^{r+1}, Y^{r+1})) - \nabla_Y f(X^r, Y^r))\| \tag{66}$$
$$\overset{(c)}{\leq} \|X^{r+1} - X^r\| + \|\nabla_X h(X^{r+1}) - \nabla_X h(X^r)\| + \sigma_Y \|\nabla_X g(X^{r+1}) - g(X^r)\|$$
$$+ \|\nabla_Y l(Y^{r+1}) - \nabla_Y l(Y^r)\|$$
$$\overset{d)}{\leq} (1 + L_X + \sigma_Y L_{g,2})\|X^{r+1} - X^r\| + (1 + L_Y) \|Y^{r+1} - Y^r\|$$

where in $(a)$ we used the optimality condition of $X^r$-subproblem; in $(b)$ we used nonexpansiveness of the projection operator; in $(c)$ we take the gradient of $f(X, Y)$; in $(d)$ we used the Lipschitz continuous of function $h(X)$ and $l(Y)$.

Since $X^r, Y^r$ are within compact sets, the sizes of $X^r, Y^r$ are bounded, which implies that there exits $\sigma_1$ such that

$$\|\nabla \mathcal{L}(X^r, Y^r)\|^2 \leq \sigma_1 (\underbrace{\|X^{r+1} - X^r\|^2 + \|Y^{r+1} - Y^r\|^2}_{\triangleq \mathcal{C}^r}). \tag{67}$$

From equation 59, we also know that there exits a constant $\sigma_2 \triangleq \min\{c_1, c_2\}$ such that

$$\mathcal{P}^r - \mathcal{P}^{r+1} \geq \sigma_2 \mathcal{C}^r. \tag{68}$$

Combining equation 67, we have

$$\|\nabla \mathcal{L}(X^r, Y^r)\|^2 \leq \frac{\sigma_1}{\sigma_2} \left( \mathcal{P}^r - \mathcal{P}^{r+1} \right). \tag{69}$$

Summing both sides of the above inequality over $r = 1, \ldots, T$, we have

$$\sum_{r=1}^{T} \|\nabla \mathcal{L}(X^r, Y^r)\|^2 \leq \frac{\sigma_1}{\sigma_2}(\mathcal{P}^1 - \mathcal{P}^{T+1}) \leq \frac{\sigma_1}{\sigma_2}(\mathcal{P}^1 - \underline{\mathcal{P}}) \tag{70}$$

where in the last inequality we have used the fact that $\mathcal{P}^r$ is decreasing and lower bounded by $\underline{\mathcal{P}}$ since $X^r, Y^r$ are within the compact sets. By utilizing the definition $T(\epsilon)$, the above inequality becomes

$$T(\epsilon)\epsilon \leq \frac{\sigma_1}{\sigma_2}(\mathcal{P}^1 - \underline{\mathcal{P}}). \tag{71}$$

Dividing both sides by $T(\epsilon)$, and by setting $C \triangleq \sigma_1/\sigma_2$, the desired result is obtained $\qquad \square$

