# OpenReview forum: "Understand the dynamics of GANs via Primal-Dual Optimization"
_ICLR.cc/2019/Conference_

### Official Review · AnonReviewer1 · 2018-10-30
**Interesting Analysis GANs' Learning Dynamics in a Limited Setting**

**Rating:** 6
**Confidence:** 3

**Review:**

This paper analyses the learning dynamics of GANs by formulating the problem as a primal-dual optimisation problem. This formulation assumes a limited class of models -- Wasserstein GANs with discriminators using linear combinations of base functions. Although this setting is limited, it advanced our understanding of a central problem related to GANs, and provides intuition for more general cases. The paper further shows the same analysis can be applied to multi-task learning and distributed learning.

Pros:

* The paper is well written and well motivated
* The theoretical analysis is solid and provide intuition for more complex problems

Cons:

* The primal-dual formulation assumes Wasserstein GANs using linear discriminator. This simplification is understandable, but it would be helpful to at least comment on more general cases.

* Experiments are limited: only results from GANs with LQG setting were presented. Since the assumption of linear discriminator (in basis) is already strong, it would be helpful to show the experimental results from this more general setting.

* The results on multi-task learning were interesting, but the advantage of optimising the mixing weights was unclear compared with the even mixture baseline. This weakens the analysis of the learning dynamics, since learning the mixing did not seem to be important.

It would also be helpful to comment on recently proposed stabilising methods. For example, would spectral normalisation bring learning dynamics closer to the assumed model?

---

> ### Author Response · Authors · 2018-11-15
> **Response of "Interesting Analysis GANs' Learning Dynamics in a Limited Setting"**
>
> Thanks sincerely for the positive feedback from the reviewer. We greatly appreciate the time and effort that have been made by the reviewer.
>
> The general GANs can be formulated as primal-dual optimization problems with both the primal and the dual are nonconvex. Proving the convergence of any algorithm on these problems is extremely challenging. To the best of our knowledge, in these general settings of GANs, the convergence of the algorithms is still an open problem. Herein, we made a reasonable assumption on the structure of the discriminator and obtained significant results under this assumption.
> The proof we have now doesn’t apply to the general cases directly as our proof relies heavily on the convexity of the inner loop maximization problem. It might be possible to generalize the proof to the general case by modifying the potential functions used in the proof, but the path is not clear at this moment.
>
> We have added one more example with MNIST data (Section 4.2). From this example, we can see the algorithm converge with a proper choice of step-size. The generated samples are not as good as those generated by general GANs. This is expected due to the linear structure of the discriminator. Nevertheless, the samples are reasonable and the quality improves as we increase the number of bases of the discriminator.
>
> The focus of our paper is the dynamics of a first-order iterative algorithm on problem (4), which includes multi-task machine learning as a special case. Our intention is not to show the formulations we adopt are better. We would refer the reviewer to (Qian et al, 2018, Namkoong & Duchi 2016) for more discussions on the advantages of the problem formulations. We have moved the multi-task learning part (including both theory and numerical results) into the supplemental materials as an extension of this work.
>
> We have commented on the stability issue in the introduction part in the revised version of the paper. We don’t see any direct relation between techniques such as spectral normalization and our assumptions on the discriminator at this moment. It could be an interesting research direction to further unravel the dynamics of GANs.

---

### Official Review · AnonReviewer2 · 2018-11-02
**Interesting theory, advantage over baseline min-max algorithms unclear**

**Rating:** 5
**Confidence:** 3

**Review:**

This paper studies the convergence of a primal-dual algorithm on a certain min-max problem and experimentally shows that it works in GANs and multi-task learning.

This paper is clear and well-written. The convergence guarantee looks neat, and convergence to stationary points is a sensible thing on non convex-concave problems. I am not super familiar with the literature of saddle-point optimization and may not have a good sense about the significance of the theoretical result.

My main concern is that the assumptions in the theory are rather over-restrictive and it’s not clear what intuitions or new messages they bring in for the practice of GANs. The convergence theorem requires the maximization problem (over discriminators) to be strictly concave. On GANs, this assumption is not (near) satisfied beyond the simple case of the LQG setting (PSD quadratic discriminators). On the other hand, the experiments on GANs just seem to say that the algorithm works but not much more beyond that. There is a brief discussion about the improvement in time consumption but it doesn’t have a report a quantitative comparison in the wall time.

On multi-task learning, the proposed algorithm shows improvement over the baseline. However it is also unclear whether it is the *formulation* (12) that brings in the improvement, or it is the actual primal-dual *algorithm*. Perhaps it might be good to try gradient descent on (12) and see if it also works well.

In general, I would recommend the authors to have a more convincing demonstration of the strength of this algorithm over baseline methods on min-max problems, either theoretical or empirical.

---

> ### Author Response · Authors · 2018-11-15
> **Response of "Interesting theory, advantage over baseline min-max algorithms unclear"**
>
> We thank the reviewer’s time and effort of reading our work. We would like to emphasize that the theoretical result is one of the most important parts of this paper. To the best of our knowledge, it is the first convergence rate result about nonconvex min-max problems. Further, the convergence rate matches the rate that the ordinary gradient descent achieves for only the minimization problems.
>
> 1, We consider Wasserstein GAN or GANs with similar structure. The major assumption we made is that the discriminator is a linear combination of predefined basis functions. We agree that this assumption might be restrictive for general GANs, but in principle, any function can be approximated to an arbitrary precision with large enough bases. Further, we would like to remark that no constraint has been imposed on the generator G; it can be any general neural network. Therefore, by using a sufficiently large set of bases, this GAN model should have the capacity to generate samples in any distribution.  Once assuming the linear structure of the bases, then it is natural to add a convex regularizer, see e.g., Sanjabi et al 2018. Hence, we don’t think strict concavity is a strong assumption.
> Though our analysis doesn’t apply to the most general framework of GANs, we believe it is big step forward as it allows us to consider general generators which introduce nonconvexity in the resulting optimization problems.
> To the best of our knowledge, all the existing works (e.g., Chen et al [2018]) used the convex-concave primal-dual dynamic to interpret GANs. Under this framework, the problem is convex and the algorithm converges to the global optimal solution of the problem, which deviates from the empirical results observed in the GANs problem. The reason is that this kind of analysis omits the inherent nature of the GANs problem which is nonconvexity. From an optimization viewpoint, our paper is the first result that shows the convergence rate of the primal-dual algorithm for nonconvex min-max problems. This part is independent of applying the primal-dual algorithm in applications of GANs.
>
> The numerical results verified the effectiveness of the proposed primal-dual algorithm in the sense that the algorithm converges stably under the different size of the regularizers, and also shows that the convergence behavior of the proposed algorithm is consistent with the theoretical analysis.
>
>
> 2, The improvement is due to the formulation of the problem shown eq. 12 which is a harder problem than the traditional minimization problem with fixed weights, in the sense that problem in Eq. 12 basically minimizes the worst case of the original problem. It can also be contributed to the fact that our proposed algorithm is able to solve the formulation well. Note that it is not possible to directly apply gradient descent to solve a min-max problem, where we definitely need some ascent technique to deal with the maximization problem.
>
>
> 3, There are few works on the nonconvex min-max problems. Prior to our work, the only algorithm that can solve the nonconvex min-max problem is proposed in the reference Sanjabi et al [2018], which can be considered a baseline work. From the theoretical point of view, the primal-dual algorithm solves the problem in an alternating way rather than the baseline method which solves the dual problem up to some high accuracy and then solve the primal problem. The proposed primal-dual algorithm is a different strategy of generating the iterations compared with the baseline work, and it has a significant advantage in terms of computational consumption. In the revised version, we also added the results that compare the proposed primal-dual algorithm with the baseline method in terms of the running time, which shows that the primal-dual method has the similar performance in terms of the number of iterations as the baseline method but uses much less computational time. Please see page 7.

---

### Official Review · AnonReviewer3 · 2018-11-04

**Rating:** 4
**Confidence:** 4

**Review:**

The paper proposed a primal-dual optimization framework for GANs and multi-task learning. It also analyzes the convergence rate of models. Some results are conducted on both real and synthetic data.

Here are some concerns for the paper:

1. The idea of the model is pretty similar with Xu et al. [2018] (Training Generative Adversarial Networks via Primal-Dual Subgradient Methods: A Lagrangian Perspective on GAN), especially the primal-dual setting. The author totally ignored it.

2. The motivation of the paper is not clear. GANs and multi-task learning are two different perspectives. Which one is your focus and what is the connection between them?

3. The experimental results are not good. The convergence analysis is good. However we also need more empirical evidence to support.

---

> ### Author Response · Authors · 2018-11-15
> **Response of the official review "Understand the dynamics of GANs via Primal-Dual Optimization"**
>
> Thanks for the reviewer’s comments. The response of each point is listed below.
> 1, Thanks for bringing to our attention the related works. In Chen et al. (2018), the authors related a class of GANs to constrained convex optimization problems. More specifically, such GANs can be viewed as Lagrangian forms of these convex optimization problems. The optimization variables in their formulation are the probability density of the generator and the function values of the discriminator. Many issues like nonconvexity do not show up. This is essentially a nonparametric model, which doesn’t apply directly to cases when the discriminator and the generator are represented by parametric models. On the other hand, our analysis is carried out on the parametric models directly and we have to deal with the nonconvexity of neural networks.  We have added the comments on this issue in the revised version of the paper and the reference Chen et al (2018) has been cited as part of the literature review. Please see page 2.
>
> 2, The GANs problem motivates us to study the dynamics of solving the nonconvex min-max saddle point problem. It turns out this formulation is very general, which also covers the problem of multi-task learning models.
> The connection between these problems is that: the problems of GANs and multi-task learning can be both formulated in the form of eq.13 under some conditions. In the revised manuscript we have added a new example for GAN and moved the multi-task learning section to the supplemental material. Please see page 8.
>
> 3, The experimental results mainly show that the convergence behavior of the proposed primal-dual algorithm is consistent with the theoretical analysis. Our intention is by no means to show our algorithm generates superior samples than other methods. Instead, due to the linear features used in the discriminator, it is expected that our generated samples are going to be worse in quality for real dataset. In the revised version, we added an example with MNIST data to further support our results. Please see page 8.
> Thanks for the appreciation of our theoretical results. We would like to remark that our goal is to understand the properties of the first-order algorithm in GAN training. Our paper presents a first theoretical result in analyzing the primal-dual algorithm for the nonconvex min-max problem that appears in GAN training.

---

### Public Comment · (anonymous) · 2018-09-30
**related works**

This is a very interesting work that analyzes the convergence of GAN.  I would like to point out that the following works also consider GAN via primal-dual optimization:

Shengjia Zhao, Jiaming Song, Stefano Ermon, "The Information Autoencoding Family: A Lagrangian Perspective on Latent Variable Generative Models", UAI 2018.
Xu Chen, Jiang Wang, Hao Ge, "Training Generative Adversarial Networks via Primal-Dual Subgradient Methods: A Lagrangian Perspective on GAN", ICLR 2018.
Farzan Farnia, David Tse, "A Convex Duality Framework for GANs", NIPS 2018.

The authors are encouraged to include these latest related papers in the literature review and point out the differences/contributions of their work.

---

> ### Author Response · Authors · 2018-10-01
> **response to the comments about "related works"**
>
> Thanks for the reviewers’ comments. These references will be included in the literature review part of this paper when a revision is allowed. Generally speaking, our response is that the main contributions of the above mentioned papers are not directly related to the those of this paper.
>
> In Zhao et al. (2018), the authors unified several generative models, e.g., VAE, infoGAN, in the Lagrangian framework. The Lagrangian problem they considered is different to ours. For one thing, the dual variable in their problem is a Lagrangian multiplier, while in our problem, it is the discriminator of GAN. Besides, the focus of their paper is not the optimization algorithm. The algorithm design and convergence analysis were not mentioned much. Our main contribution, on the other hand, is a convergence proof of a first-order primal-dual algorithm for GANs.
>
> In Chen et al. (2018), the authors related a class of GANs to constrained convex optimization problems. More specifically, such GANs can be viewed as Lagrangian forms of these convex optimization problems. The optimization variables in their formulation are the probability density of the generator and the function values of the discriminator. Many issues like nonconvexity do not show up. This is essentially a nonparametric model, which doesn’t apply to cases when the discriminator and the generator are represented by parametric models. On the other hand, our analysis is carried out on the parametric models directly and we have to deal with the nonconvexity of neural networks.
>
> We couldn’t find the preprint of Farnia et al. (2018). Based on the abstract on the NIPS website, we believe the primal-dual formulations we considered are similar in the sense that the discriminator is constrained to a convex set. However, the focus of Farnia et al. (2018) is not the convergence properties of optimization algorithms, instead, they investigated the properties of the optimal solutions. We think our convergence analysis of the first-order primal-dual algorithm is complementary to their results.

---

> > ### Public Comment · (anonymous) · 2018-10-27
> > **Related work**
> >
> > This work is also related to the following paper, which uses prediction steps (which can be primal-dual optimization with extra gradients):
> >
> > Abhay Yadav, Sohil Shah, Zheng Xu, David Jacobs, Tom Goldstein, "Stabilizing Adversarial Nets with Prediction Methods", ICLR 2018.
> >
> > The authors are encouraged to cite and discuss the differences/contribution of their work.

---

> > > ### Author Response · Authors · 2018-11-15
> > > **The related work is included**
> > >
> > > Thanks for the comment. Yadav et al. (2018) considered convex-concave primal-dual optimization problems. This is considerably different to our setup where GANs, as they should be, are formulated as nonconvex saddle point problems. We have included this reference in the revised version of this paper.

---

### Meta-Review · Area_Chair1 · 2018-12-16
**A primal-dual algorithm for linear discriminator WGANs with first order convergence, as a special non-convex optimization problem.**

**Confidence:** 4
**Recommendation:** Reject

**Metareview:**

The paper studies the convergence of a primal-dual algorithm on a special min-max problem in WGAN where the maximization is with respect to linear variables (linear discriminator) and minimization is over non-convex generators. Experiments with both simulated and real world data are conducted to show that the algorithm works for WGANs and multi-task learning.

The major concern of reviewers lies in that the linear discriminator assumption in WGAN is too restrictive to general non-convex mini-max saddle point problem in GANs. Linear discriminator implies that the maximization part in min-max problem is concave, and it is thus not surprise that under this assumption the paper converts the original problem to a non-convex optimization instance and proves its first order convergence with descent lemma. This technique however can’t be applied to general non-convex saddle point problem in GANs. Also the experimental studies are also not strong enough. Therefore, current version of the paper is proposed as borderline lean reject.